# Differential involvement of feedback and feedforward control networks across disfluency types in adults who stutter: Evidence from resting state functional connectivity

Hannah P. Rowe[1,2]*, Saul A. Frankford[2,3], Jackie S. Kim[2], Jason A. Tourville[2], Alfonso Nieto-Castanon[2], Frank H. Guenther[2,4,5,6]

1 Department of Communication Sciences and Disorders, Northeastern University, Boston, Massachusetts, United States of America, 2 Department of Speech, Language and Hearing Sciences, Boston University, Boston, Massachusetts, United States of America, 3 Department of Speech, Language, and Hearing, University of Texas Dallas, Dallas, Texas, United States of America, 4 Department of Biomedical Engineering, Boston University, Boston, Massachusetts, United States of America, 5 MIT, Picower Institute for Learning and Memory, Cambridge, Massachusetts, United States of America, 6 Harvard/MIT, Speech and Hearing Bioscience and Technology Program, Cambridge, Massachusetts, United States of America

* h.rowe@northeastern.edu

## Abstract

### Purpose

This study investigated the relationship between different disfluency types (i.e., repetitions, prolongations, and blocks) and resting state functional connectivity in the feedback (FB) and feedforward (FF) control networks in 20 adults who stutter.

### Methods

Frequency of each disfluency type was coded in speech samples derived from the Stuttering Severity Instrument, and functional connectivity between brain regions of interest was derived from resting state functional magnetic resonance imaging scans. We used LASSO regressions to identify the connections that most strongly predicted each disfluency type.

### Results

Both repetitions and prolongations were significantly associated with increased connectivity in *left ventral motor cortex – right ventral premotor cortex*, which is hypothesized to be involved in FB control of speech. In contrast, blocks were significantly associated with reduced connectivity in *right anterior cerebellum – left ventral lateral thalamic nucleus* and increased connectivity in *left presupplementary motor area – left posterior inferior frontal sulcus*, both of which are hypothesized to be involved in FF control of speech.

**Data availability statement:** The data underlying the results presented in the study are available on Open Source Foundation at: https://osf.io/t5k74/.

**Funding:** This work was supported by the National Institutes of Health (National Institute of Deafness and Other Communication Disorders, https://www.nidcd.nih.gov/) [FHG: R01DC007683 and HPR: T32DC013017-07 ]. The funders had no role in study design, data collection and analysis, decision to publish, or preparation of the manuscript.

**Competing interests:** Frank Guenther receives royalties for his book Neural Control of Speech from MIT Press. All other authors have no known competing financial interests or personal relationships that could have appeared to influence the work reported in this paper. This does not alter our adherence to PLOS ONE policies on sharing data and materials.

## Conclusion

Our findings suggest that repetitions and prolongations may be associated with increased reliance on FB-based corrective mechanisms, whereas blocks may be associated with disrupted FF-based initiation mechanisms. These neural underpinnings may correspond to different challenges in terminating or initiating motor commands and underscore the nuanced neurobiological processes underlying speech disfluencies.

## Introduction

More than 50 million adults worldwide live with persistent developmental stuttering [1,2], which can have profound adverse effects on quality of life [3]. Stuttering is generally characterized by overt speech behaviors such as repetitions (i.e., repeated sounds or syllables), prolongations (i.e., extended duration of a sound), and blocks (i.e., temporary stoppage of airflow or vocalization). However, the prevalence of each type can vary widely across individuals. Prior research has highlighted individual differences and potential subgroups in the expression of disfluencies among adults and children who stutter [4–8]. This heterogeneity presents challenges in identifying the underlying neural mechanisms of stuttering [9–12]; neuroimaging studies have struggled to offer a unified understanding of the brain regions involved, possibly because key insights are obscured by grouping all individuals who stutter together despite differences in underlying etiologies. A clearer mechanistic understanding of how specific neural control systems contribute to individual stuttering patterns could inform the development of more targeted, individualized treatment strategies, potentially improving therapy outcomes for adults who stutter (***AWS***) [13–19].

### The role of feedback and feedforward control in stuttering

Over the past several decades, a substantial number of neuroimaging studies have expanded our understanding of the neural mechanisms underlying stuttering. Researchers have found associations between disfluency rates and functional and structural anomalies in left inferior frontal gyrus, bilateral temporal regions, right frontal regions, and speech motor networks involving basal ganglia, thalamus, and sensorimotor cortex (see [9,20–22] for reviews). Increased cerebellar activity has also been consistently observed across studies, with many researchers arguing that it reflects a compensatory mechanism for timing deficits in AWS [23–31].

Two mechanisms that have garnered increasing interest for understanding stuttering are feedback (***FB***) control (i.e., integrating information to monitor and correct for errors) and feedforward (***FF***) control (i.e., adjusting the system through learning to preemptively avoid producing errors) of speech production. These control processes have been associated with distinct brain networks, as detailed in the Directions into the Velocities of Articulators (DIVA) model [32] and Gradient Order DIVA (GODIVA) model [33], which are theoretical frameworks that offer comprehensive accounts of the neural mechanisms involved in speech motor control. The DIVA model proposes

that speech production relies on a combination of FB and FF control processes, highlighting the role of the cerebellum in fine-tuning motor plans and the basal ganglia in regulating the initiation of speech. The GODIVA model builds upon DIVA by further specifying the gradient organization of motor planning, with a focus on how complex articulatory movements are organized hierarchically across different levels of speech production. Together, these models offer valuable insight into how disruptions in the FB and FF systems could contribute to disfluencies like stuttering, particularly in the coordination of speech movements and error correction.

Over the years, there has been an increased focus on FB and FF control as a means of understanding the behavioral and neural findings associated with stuttering. Several leading etiological theories have implicated FB control in the cause of stuttering, suggesting that difficulties detecting errors and making adjustments during speech can lead to disfluencies [34–36]. These theories are supported by evidence from behavioral and neuroimaging studies, which reveal differences in FB-based processing [37–45], such as greater reliance on auditory FB [46–48], and anomalies in planum temporale and middle temporal gyrus, areas thought to contribute to FB control [18,49–51]. Other theories have proposed that deficient FF control is the primary cause of stuttering, suggesting that difficulties planning and/or initiating movements are at the core of disfluencies [9]. Consistent with these theories, behavioral and neuroimaging studies have revealed differences in FF-based processing [9,34–36,38–40,43,45] and anomalies in brain regions hypothesized to contribute to FF control [50,52,53].

A common theme across this literature is that individuals who stutter are not a homogeneous group but rather exhibit diverse neural profiles [9,10,13,18]. Indeed, in many of the studies mentioned above, researchers found that only a subset of AWS relied more on auditory FB [48] or exhibited strengthened FB or FF network connectivity following treatment [50]. Researchers have also identified subgroups based on atypical symmetry of planum temporale [18], which is hypothesized to be associated with FB processing, and atypical caudate anatomy [52] and underactivity in basal ganglia pathways [9,13], both of which are hypothesized to be associated with FF processing. This within-population variability suggests that FB and FF control may influence stuttering to varying degrees across different individuals who stutter.

## Prior literature on neural underpinnings of specific disfluency types

Despite the heterogeneity in stuttering types and in the brain mechanisms involved in stuttering across AWS, no study to our knowledge has specifically examined the neural underpinnings of the three primary disfluency types. One prior study used pattern classification to automatically categorize disfluencies that are more typical (**MT**) of stuttering (i.e., part-word repetitions, prolongations, and broken words) and less typical (**LT**) of stuttering (i.e., incomplete phrases, revisions, interjections, and phrase repetitions) in AWS and neurotypical speakers using corresponding task-based fMRI data [54]. The authors found that left inferior frontal cortex and bilateral precuneus showed higher activity for MT disfluencies, while left putamen and right cerebellum showed higher activity for LT disfluencies. These results demonstrate that different clusters of disfluencies (i.e., MT versus LT) can be attributed to differences in brain activity, but they address the neural underpinnings of MT disfluencies as a whole rather than of the individual disfluency types.

While less direct than studies examining human behavior, computational modeling can be used to provide insights into the neural processes underlying speech. The DIVA and GODIVA models, specifically, have been used to model the impact of brain function hypothesized to be involved in FB and FF control. Over five decades ago, a series of seminal studies provided evidence proposing that repetitions may be related to FB control, specifically; these studies demonstrated an absence/reduction of repetitions when auditory FB was not available due to masking [55–58]. Based on this work, DIVA model simulations tested the association between FB control and repetitions and demonstrated that overreliance on FB control (and thus oversensitivity to speech errors) could lead to repeated or prolonged phonemes [59]. By increasing the gain on the FB control system, the simulations induced repetition errors that were acoustically and kinematically similar to those of AWS in prior behavioral studies [60–64]. In another simulation, masking noise was added to the auditory FB signal and successfully reduced repetitions, indicating that noise prevents detection (and subsequent overcorrection) of errors, consistent with the aforementioned behavioral studies in AWS [65,66].

Neural modeling has also provided evidence of the potential contribution of FF control to the production of blocks. GODIVA simulations investigated the hypothesis that elevated dopamine levels in a basal ganglia thalamo-cortical (BGTC) circuit could lead to a failure to initiate the upcoming syllable in a timely manner, suggesting that blocks could arise from impairments in FF control [67,68]. The authors mimicked the effects of excess dopamine in the system by artificially reducing the signal to noise ratio between the motor program for the upcoming syllable and those for the other syllables. This manipulation resulted in the model's inability to bias the upcoming syllable for selection, ultimately causing a block in speech production. These findings underscore the possible role of the FF control network in the timely initiation of speech sounds.

### The current study

The simulation experiments discussed above illustrate that targeted disruptions to the FB and FF networks for speech can lead to distinct stuttering behaviors [59,67,68]. This insight motivated the current study, which investigates how feedforward (FF) and feedback (FB) control networks relate to different stuttering types by examining correlations between disfluency patterns and resting-state connectivity in individuals with persistent stuttering. While the aforementioned studies primarily focused on task-based fMRI to examine neural mechanisms underlying speech production, resting-state functional connectivity offers complementary insights by capturing a stable measure of functional network organization, reflecting the brain's inherent activity patterns and neural traits that persist across contexts. This approach allows us to investigate baseline connectivity patterns that may underlie susceptibility to disfluencies, rather than neural responses constrained by specific task performance. Furthermore, resting-state functional connectivity analysis circumvents challenges commonly encountered in task-based studies of stuttering, such as difficulty in eliciting spontaneous disfluencies under controlled experimental conditions and increased motion artifacts during speech production tasks. Prior work has also demonstrated strong correlations between resting state functional connectivity and specific behaviors (e.g., inhibitory control in attention-deficit/hyperactivity disorder) [69] and its ability to distinguish subtypes of social-cognitive and neuro-cognitive differences in clinical populations [70,71].

Based on the aforementioned literature [55–59,67,68], we hypothesize that repetitions and prolongations will be primarily associated with *increased* connectivity in the FB control network, reflecting an overreliance on FB mechanisms, while blocks will be primarily associated with *reduced* connectivity in the FF control network, reflecting weakened FF mechanisms. Notably, although the DIVA model does not explicitly define this distinction, we divided Cb into anterior and posterior regions to reflect functional differences; anterior regions are primarily involved in motor aspects of speech, while posterior regions are associated with higher-level cognitive functions [72,73].

Our goal is to explore potential associations that could guide future research into more targeted and individualized models of stuttering. For example, by identifying how specific neural control systems contribute to different disfluency types, we may eventually be able to tailor therapies that target the dominant mechanisms affecting each individual. However, we do not aim to make definitive conclusions at this stage, but rather to identify trends that might warrant further investigation in subsequent studies. Finally, while in this study we focused on overt speech disruptions, we acknowledge that these behaviors do not capture the full complexity of the stuttering experience. However, focusing on these observable features provides a quantifiable and reproducible starting point for identifying potential neural mechanisms, which can later be expanded to encompass the broader phenomenology of stuttering.

## Materials and methods

### Participants

Participants included 20 AWS (15 male, 5 female; aged 18−52 years, mean age = 30.50 +/- 11.89 years). All participants were diagnosed with stuttering based on self-identification as a person who stutters during a screening interview and

self-report that certain sounds and social contexts impact their overt symptoms. The ratio of males to females in this study is consistent with the prevalence of persistent developmental stuttering in the national population [74]. All participants were native English speakers, reported normal hearing and vision, had no prior history of speech, language, or hearing disorders other than stuttering, and self-identified as stuttering during a screening interview. Using the Edinburgh Handedness Inventory [75], all participants were found to be right-handed (i.e., scoring greater than 40). The study was approved by the Boston University Institutional Review Board (#2421) and all participants provided written informed consent. Data were accessed on November 9, 2023 for research purposes. The author conducting the analyses did not have access to information that could identify individual participants. See Table 1 for demographic information.

## Behavioral data acquisition

Stuttering severity and disfluency types (i.e., repetitions, prolongations, and blocks) were derived from the Stuttering Severity Instrument – Fourth Edition (SSI-4 [76];), which included three speech samples: an in-person conversation, a phone conversation, and the Grandfather reading passage [77]. The topic of the in-person conversational samples varied across participants, with examples such as their current job, hobbies, or future vacation plans, and they were 4–6 minutes and 415 syllables long, on average. For the reading sample, each participant was told to: "Please read this paragraph. Begin when you are ready." Each reading sample was 2–3 minutes and 170 syllables long, on average. Lastly, the phone call samples were 2–3 minutes and 200 syllables long, on average. The frequency of each stuttering type was calculated as the average across all three speech samples, giving equal consideration to each task. All speech samples were audio

**Table 1. Demographic information and stuttering behavior data of adults who stutter. Percentages are calculated by dividing number of repetitions, prolongations, or blocks by total number of syllables produced.**

| Participant ID | Age | Gender | Scanner | SSI-4 (range 9–42) | Total Number of Syllables | Repetition Raw Count and Percentage | Prolongation Raw Count and Percentage | Block Raw Count and Percentage |
|---|---|---|---|---|---|---|---|---|
| AWS01 | 19 | F | MGH | 28 | 791 | 11 (1.39%) | 0 (0%) | 54 (6.83%) |
| AWS02 | 22 | F | MGH | 31 | 542 | 18 (3.32%) | 11 (2.03%) | 84 (15.50%) |
| AWS03 | 31 | F | MGH | 10 | 704 | 3 (0.43%) | 0 (0%) | 9 (1.28%) |
| AWS04 | 31 | F | MGH | 30 | 737 | 6 (0.81%) | 4 (0.54%) | 45 (6.11%) |
| AWS05 | 18 | F | BU | 14 | 770 | 1 (0.13%) | 0 (0%) | 27 (3.51%) |
| AWS06 | 23 | M | MGH | 20 | 868 | 12 (1.38%) | 5 (0.58%) | 24 (2.77%) |
| AWS07 | 19 | M | MGH | 9 | 346 | 2 (0.58%) | 0 (0%) | 4 (1.16%) |
| AWS08 | 29 | M | MGH | 33 | 880 | 24 (2.73%) | 1 (0.11%) | 97 (11.02%) |
| AWS09 | 58 | M | MGH | 14 | 857 | 21 (2.45%) | 1 (0.12%) | 6 (0.70%) |
| AWS10 | 42 | M | MGH | 22 | 1069 | 21 (1.96%) | 1 (0.09%) | 13 (1.22%) |
| AWS11 | 22 | M | MGH | 42 | 248 | 4 (1.61%) | 3 (1.21%) | 24 (9.68%) |
| AWS12 | 52 | M | MGH | 27 | 698 | 10 (1.43%) | 0 (0%) | 29 (4.15%) |
| AWS13 | 44 | M | BU | 20 | 793 | 12 (1.51%) | 4 (0.50%) | 18 (2.27%) |
| AWS14 | 29 | M | BU | 14 | 847 | 16 (1.89%) | 9 (1.06%) | 15 (1.77%) |
| AWS15 | 20 | M | BU | 18 | 799 | 1 (0.12%) | 6 (0.75%) | 18 (2.25%) |
| AWS16 | 43 | M | BU | 27 | 913 | 49 (5.37%) | 17 (1.86%) | 15 (1.64%) |
| AWS17 | 35 | M | BU | 30 | 814 | 11 (1.35%) | 8 (0.98%) | 43 (5.28%) |
| AWS18 | 22 | M | BU | 27 | 759 | 10 (1.32%) | 1 (0.13%) | 28 (3.69%) |
| AWS19 | 21 | M | BU | 19 | 827 | 14 (1.69%) | 7 (0.85%) | 33 (3.99%) |
| AWS20 | 20 | M | BU | 24 | 812 | 1 (0.12%) | 1 (0.12%) | 33 (4.06%) |

SSI-4 = Stuttering Severity Index–Fourth Edition; AWS = adults who stutter; F = female; M = male; MGH = Massachusetts General Hospital; BU = Boston University.

and video recorded. For each sample, a licensed speech-language pathologist who was board certified in fluency calculated the number of total syllables produced, total stuttered syllables, and total number of each disfluency type.

Stuttering behaviors were operationally defined based on well-established criteria from the literature [78]. The three primary disfluency types—repetitions, prolongations, and blocks—were categorized according to standardized definitions: repetitions refer to the involuntary repetition of sounds, syllables, or words; prolongations refer to the extended duration of speech sounds; and blocks refer to interruptions in the flow of speech, often accompanied by physical tension. These behaviors were reliably identified and scored using established coding guidelines to ensure consistency across participants. When mixed stuttering was present, which involves the occurrence of two or more different stuttering types in a single stuttering moment (e.g., a repetition combined with a prolongation or block), the stutter was classified according to the dominant disfluency type within that occurrence. "Dominance" was operationalized as the feature occupying the greater proportion of the stuttering moment. In rare cases where no type was clearly dominant, the classification was determined based on the first disfluency to occur in the sequence. Because blocks are often perceptually salient, coders were trained to apply this rule conservatively to avoid over-classification of blocks when they occurred in combination with repetitions or prolongations. This approach ensured that mixed disfluencies were consistently classified while minimizing the risk that blocks would be disproportionately coded as dominant. To confirm reliability, inter-rater agreement between three separate raters (i.e., the original rater and two additional speech-language pathologists) was assessed on 20% of the data. This analysis demonstrated an ICC of.96 for repetitions,.80 for prolongations, and.76 for blocks, all of which are within the range of good to excellent reliability. See Table 1 for stuttering severity and stuttering type data.

### Resting state data acquisition

MRI data were acquired approximately two months, on average, after the behavioral data acquisition. Data were acquired at two sites: the Athinoula A. Martinos Center for Biomedical Imaging at the Massachusetts General Hospital (MGH) in Charlestown, MA and the Cognitive Neuroimaging Center at Boston University (BU). Images were acquired using a Siemens 3T MAGNETOM Skyra scanner with a 32-channel head coil at MGH and a Siemens 3T Prisma scanner with a 64-channel head coil at BU. To acquire resting state functional connectivity data, participants were positioned supine in the scanner for six minutes. They were given the following instructions: "Please lie quietly with your eyes open and observe the cross. During this scan you should let your mind wander freely. Please remain awake and keep still during this scan. This scan will last six minutes." A gradient echo, echo-planar imaging BOLD sequence (repetition time [TR] = 1130 ms, echo time = 30 ms, flip angle = 60°, simultaneous multi-slice factor = 2) was used to collect 315 continuous volumes. Each functional volume comprised 51 axial slices (72 × 72 matrix) acquired in interleaved order (in-plane resolution = 2.97 x 2.97 mm²; slice thickness = 3.0 mm with no gap). A high-resolution T1-weighted whole-brain structural image was also collected from each participant to anatomically localize the functional data (MPRAGE [magnetization-prepared rapid gradient-echo] sequence; 256 × 256 × 176 mm³ volume, with a 1-mm isotropic resolution; TR = 2530 ms; echo time = 1.69 ms; flip angle = 7°).

### Functional connectivity data analysis

**Preprocessing.** Preprocessing was conducted using the CONN toolbox [79], which is one of the most widely used tools for resting-state fMRI analyses, as it provides a robust framework for functional connectivity analyses and has been validated in the field [80,81]. Two distinct pipelines were employed: a surface-based approach to extract cortical activation and a volume-based approach to extract subcortical and cerebellar activation. Our decision to combine surface-based methods for cortical regions and volumetric approaches for subcortical areas is guided by previous work demonstrating that surface-based analysis provides superior alignment of cortical folding patterns [82–84]. In contrast, volumetric approaches are recommended for subcortical structures, where complex three-dimensional architecture cannot

be effectively captured by surface-based methods [85]. Using both approaches leverages the unique strengths of each method, maximizing anatomical accuracy across all relevant brain regions.

Preprocessing common to both pipelines included slice time correction, motion correction, and outlier detection, followed by denoising with aCompCor [86]. Using Statistical Parametric Mapping (SPM12)'s realign and unwarp procedure, functional images were adjusted for any misalignments based on the mean subject image and corrected for motion-by-inhomogeneity interactions [87]. Artifact detection tools (https://www.nitrc.org/projects/artifact_ detect/) identified outlier scans based on motion displacement (scan-to-scan motion threshold of 0.9 mm) and mean signal change (scan-to-scan signal change threshold of 5 SDs above the mean). To remove any extraneous motion, physiological, and artifactual effects from the BOLD signal for each subject, denoising was performed using a linear regression model in each seed ROI and every voxel in the smoothed brain volume, including five white matter regressors, five cerebrospinal fluid regressors, six subject motion parameters, and scrubbing regressors to exclude the effects of outlier scans, as well as individual regressors for each run and session, task condition, and error trial [88].

For surface-based analyses, FreeSurfer [83] was employed to preprocess the T1 structural volumes by removing non-brain components [89] and segmenting the brain into gray matter, white matter, and cerebrospinal fluid components [90]. Cortical surface reconstructions for each hemisphere were then generated [91] and registered to a standard surface template (*fsaverage*). The resting state fMRI volume was co-registered to the structural volume to align the functional and anatomical data. Blood oxygen level dependent (BOLD) responses from the resting state fMRI data were then mapped to their corresponding vertices on the cortical surface. The surface-based representation of the resting state fMRI data was smoothed using iterative diffusion smoothing with 40 diffusion steps (equivalent to an 8-mm full-width half-maximum smoothing kernel; [92]) for vertex-level analyses.

For volume-based analyses, functional volumes were co-registered to the T1 structural images and then segmented and normalized to Montréal Neurological Institute (MNI) space after outlier detection using SPM12 [93]. The structural T1 images were centered, segmented, and normalized with SPM12 to ensure accurate alignment with the functional data. Following co-registration, volumes were smoothed with an 8-mm full-width half-maximum smoothing kernel for voxel-level analyses. Afterward, a mask was applied to include only voxels within subcortical structures for subsequent analyses. After preprocessing, two AWS participants were excluded from further analyses (not included in the *Subjects* section): one due to excessive head motion in the scanner (> 1.5 mm average scan-to-scan motion) and the other due to structural brain abnormalities.

**Region-of-interest definition.** Cortical regions-of-interest (**ROIs**) were identified using a modified version of the SpeechLab atlas, as described in Cai and colleagues (2014). This atlas divides the cortex into macro-anatomically defined ROIs designed specifically for speech studies. The labeling was performed by mapping the atlas from the FreeSurfer *fsaverage* cortical surface template onto each individual's surface reconstruction.

Subcortical and cerebellar ROIs were sourced from multiple different atlases. Thalamic ROIs were obtained from the mean atlas of thalamic nuclei [94]. Basal ganglia ROIs were extracted from the nonlinear normalized probabilistic atlas of basal ganglia [95]. Each ROI was thresholded at a minimum probability of 33% and combined into a single labeled volume within the atlas's native space (the MNI104 template). Cerebellar ROIs were extracted from the cerebellar networks included in the CONN toolbox. Each atlas was nonlinearly registered to the SPM12 MNI152 template and then merged into a single labeled volume.

**Region-of-interest-based connectivity analyses.** ROI-based connectivity analyses were performed using the CONN toolbox, which computed the Fisher-transformed temporal correlation between ROI pairs. ROI-to-ROI correlations were then extracted for connections described in the DIVA and GODIVA models' FB and FF control networks (including anterior and posterior Cb). Our focus on DIVA-based connections was a strategic choice to: (1) maximize the power of the analysis given our small sample size, and (2) directly test hypotheses derived from the model, ensuring the analysis was grounded in established theoretical and empirical research. By concentrating on a targeted set of connections

directly relevant to the speech production process, we aimed to reduce the risk of Type II errors and maintain sufficient sensitivity to detect meaningful effects. To reduce the total number of connections analyzed, we focused on left-lateralized connections and eliminated right hemisphere connections that were functionally redundant with those in the left, retaining only select right hemisphere connections (i.e., connections to right ventral premotor cortex and right cerebellum). In light of findings of increased right hemisphere involvement in stuttering, we also included connections with right brain regions that have been consistently found to be active in prior literature in AWS (see 9,20,22 for reviews). In the end, a total of 58 Fisher-transformed correlation coefficients were included as predictors in the subsequent statistical analyses.

## Statistical analyses

Statistical analyses were performed using R. To identify the connections that most strongly predicted each disfluency type, we created 1000 bootstrapped samples through random sampling with replacement from the original dataset, generating multiple resampled datasets to assess the stability of our estimates. Both the predictors (i.e., the DIVA/GODIVA connections) and the response variables (i.e., repetitions, prolongations, and blocks) were bootstrapped, ensuring a comprehensive evaluation of the robustness of our model. Bootstrapping was applied to mitigate potential overfitting and provide more reliable estimates of the model coefficients [96]. This approach allowed us to assess the stability of the relationships between predictors and response variables, improving the generalizability and confidence in our findings.

For each bootstrapped sample, we conducted three Least Absolute Shrinkage and Selection Operator (LASSO) regressions—one for each disfluency type—using the *cv.glmnet* function from the *glmnet* package. The predictors included the 58 aforementioned Fisher-transformed correlation coefficients, while the outcome was percentage of each disfluency type across all syllables in the speech sample. To control for potential scanner-related effects, the scanner variable was included in each model, with binary encoding (1 for MGH and 0 for BU). Statistical control of scanner effects accounts for differences in average connectivity due to acquisition parameter variability across sites. We also conducted Levene's tests to assess potential heteroscedasticity introduced by inter-site differences in variance. The results did not indicate a significant effect (see Table S1 in Supplementary Material), suggesting that further normalization across sites was not necessary. The LASSO models were then fitted with the optimal lambda (regularization parameter) determined via cross-validation. LASSO regression prevents overfitting when working with small, high-dimensionality datasets, and is one of the most commonly used penalized regression methods [97].

Next, we calculated the average coefficient values and standard deviation across the 1000 iterations for each predictor. To assess whether each coefficient value was statistically significantly different from zero, we constructed 95% confidence intervals (CIs) for each raw LASSO coefficient using the standard errors derived from the bootstrapped samples. A predictor was considered statistically significant if its 95% CI did not include zero, indicating a consistent, non-random effect across the resampled datasets. To account for multiple comparisons, we applied a Bonferroni correction, adjusting the threshold for statistical significance. This bootstrapped CI approach provides a robust measure of significance, particularly well-suited for high-dimensional data, as it accounts for variability in the estimates without relying on strict parametric assumptions.

To ensure the robustness of our coefficient estimates, we applied a stability selection method [98], which involved calculating the frequency of non-zero coefficients for each predictor across the 1000 iterations. This method helped to identify which predictors were consistently influential. To determine a cutoff for variable inclusion based on stability selection frequencies, we used a data-driven heuristic inspired by scree plot and elbow methods. Specifically, we identified the index at which the first prominent drop in selection frequency occurred among predictors, ranked by their selection count across bootstrap samples. This point was interpreted as the last relatively stable variable before the onset of a sharp decline, indicating a transition from highly stable to less consistently selected predictors. Although this approach does not provide formal control of false discoveries, it offers a pragmatic balance between sparsity and reproducibility in the context of exploratory feature selection. Compared to traditional elbow methods, our "first drop" heuristic imposes a more conservative criterion for feature selection. Rather than identifying the point of diminishing returns, it targets the onset of

instability in selection frequencies, yielding a smaller but more stable set of predictors. This stricter threshold prioritizes reproducibility over model complexity, aligning with our goal of identifying robust, high-confidence variables.

## Results

### Behavioral analysis

In our study of 20 participants, the mean SSI-4 score was 22.95 (SD = 8.48), indicating mild to moderate severity across the sample. On average, repetitions made up 1.21% (SD = 1.14) of all syllables spoken, prolongations made up 0.47% (SD = 0.57), and blocks made up 3.72% (SD = 4.47). Repetitions and prolongations comprised a relatively small proportion of the syllables spoken, but there was considerable variability among participants, with some showing minimal to no repetitions/prolongations and others displaying more frequent occurrences. Blocks were more common, making up a larger proportion of the total syllables produced compared to repetitions and prolongations. However, the large standard deviation also suggests considerable variability in the frequency of blocks.

### Brain-behavior analysis

The LASSO regression analysis aimed to identify resting state functional connectivity patterns underlying different disfluency types. After conducting bootstrapping with 1000 resampled datasets, LASSO regressions were performed for each disfluency type. Optimal lambda values, which control the strength of the regularization, were determined through cross-validation, minimizing the mean squared error. These optimal lambda values were then used to fit the LASSO models, resulting in a mean coefficient value (also referred to as "estimate" below) across the 1000 iterations for each regression. For repetitions, prolongations, and blocks, 27, 26, and 23 predictors, respectively, showed statistically significant effects. In each case, the confidence intervals did not include zero, indicating consistent and non-random relationships across the resampled datasets (see Table S2 in Supplementary Material). Notably, although blocks were associated with fewer overall predictors than repetitions or prolongations, the predictors that did emerge for blocks showed larger effect estimates and required greater regularization, suggesting stronger and potentially more robust brain–behavior relationships. By contrast, prolongations exhibited fewer and weaker associations, consistent with their relatively lower prevalence in the behavioral data. These results, derived from optimal lambda values determined via cross-validation, demonstrate robust and significant functional connectivity patterns related to disfluency types. See Fig 1 for a heatmap of LASSO regression coefficients from the most optimal model across the bootstrapped iterations.

The stability selection method was then conducted to identify the most consistently influential predictors across the 1000 iterations. Lastly, a data-driven cutoff was identified based on the point just before the largest drop in selection frequency, capturing the most consistently selected predictors across model iterations (see Fig S1 in Supplementary Material). The following connections were the most consistent across iterations (outlined in black rectangles in Fig 1): Repetitions (average optimal λ = 0.09) were primarily predicted by increased connectivity between *left ventral motor cortex (vMC)* and *right ventral premotor cortex (vPMC)* (FB control network; estimate = 0.21, SE = 0.008); prolongations (average optimal λ = 0.11) were primarily predicted by increased connectivity between *left ventral motor cortex (vMC)* and *right ventral premotor cortex (vPMC)* (FB control network; estimate = 0.04, SE = 0.003); and blocks (average optimal λ = 0.75) were primarily predicted by reduced connectivity between *right anterior cerebellum (aCb)* and *left ventral lateral thalamic nucleus (VL)* (FF control network; estimate = −0.18, SE = 0.015) and increased connectivity between *left presupplementary motor cortex (preSMA)* and *left posterior inferior frontal sulcus (pIFS)* (FF control network; estimate = 0.17, SE = 0.014). See Fig 2 for the correlation plots of each disfluency type and area of functional connectivity.

Lambda values indicate the degree of regularization applied in the LASSO regression models. Lower lambda values for repetitions and prolongations suggest that these types of disfluencies are more sensitive to subtle variations in connectivity. Conversely, a higher lambda for blocks implies that more regularization was needed, indicating potentially greater complexity or variability in the predictors. For repetitions, the relatively higher estimate (0.21) with a lower standard error

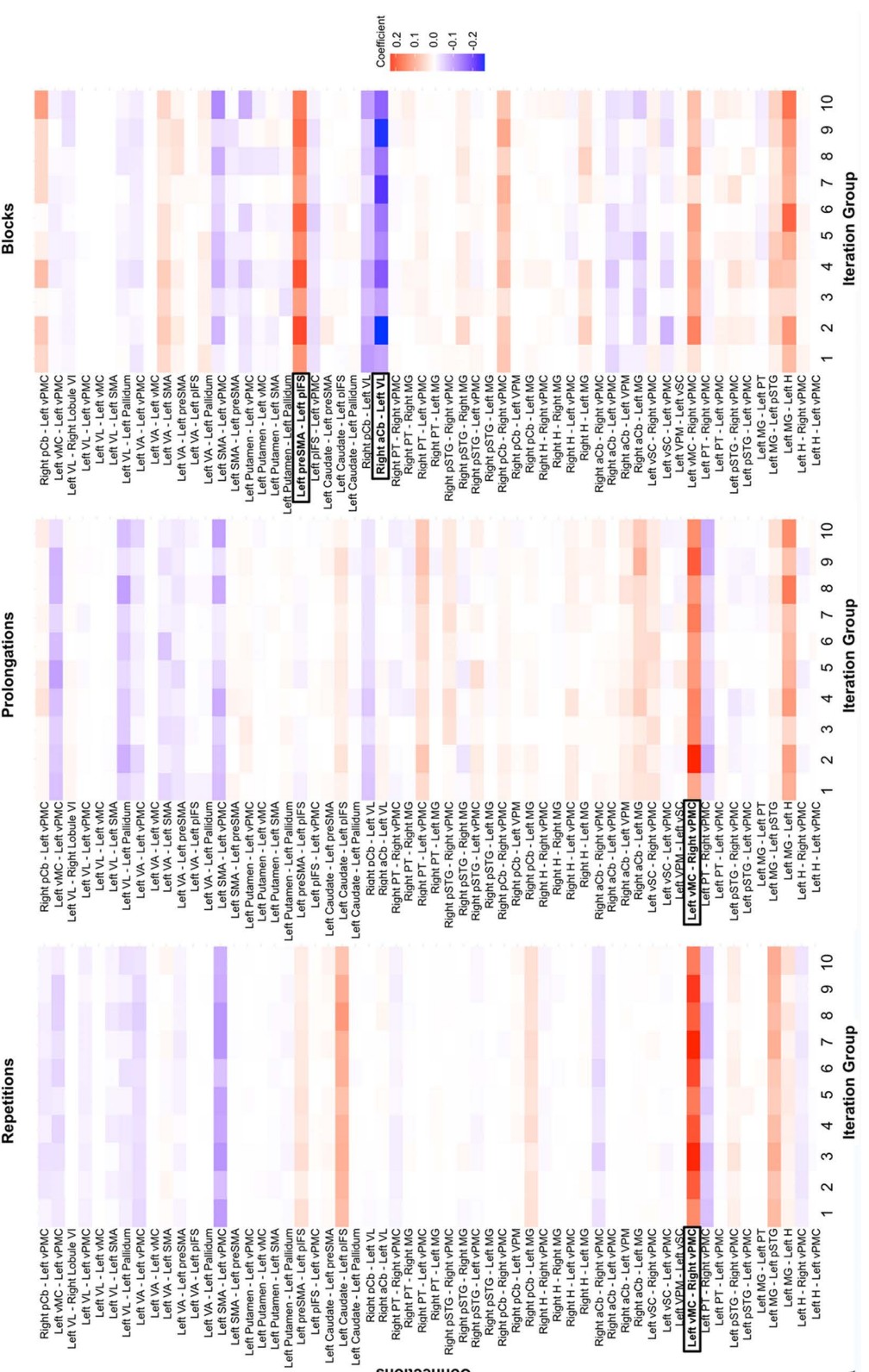

**Fig 1. Heatmap of LASSO regression coefficients averaged across 1000 bootstrapped iterations.** Color intensity = coefficient magnitude; black rectangles = most consistently selected predictors; a/pCb = anterior/posterior cerebellum; GP = globus pallidus; H = Heschl's gyrus; MG = medial geniculate thalamic nucleus; pAC = posterior auditory cortex; pIFS = posterior inferior frontal sulcus; preSMA = presupplementary motor area; pSTG = posterior superior temporal cortex; PT = planum temporale; SMA = supplementary motor area; VA = ventral anterior thalamic nucleus; VL = ventral lateral thalamic nucleus; vMC = ventral motor cortex; vPMC = ventral premotor cortex; VPM = ventral posterior medial thalamic nucleus; vSC = ventral somatosensory cortex.

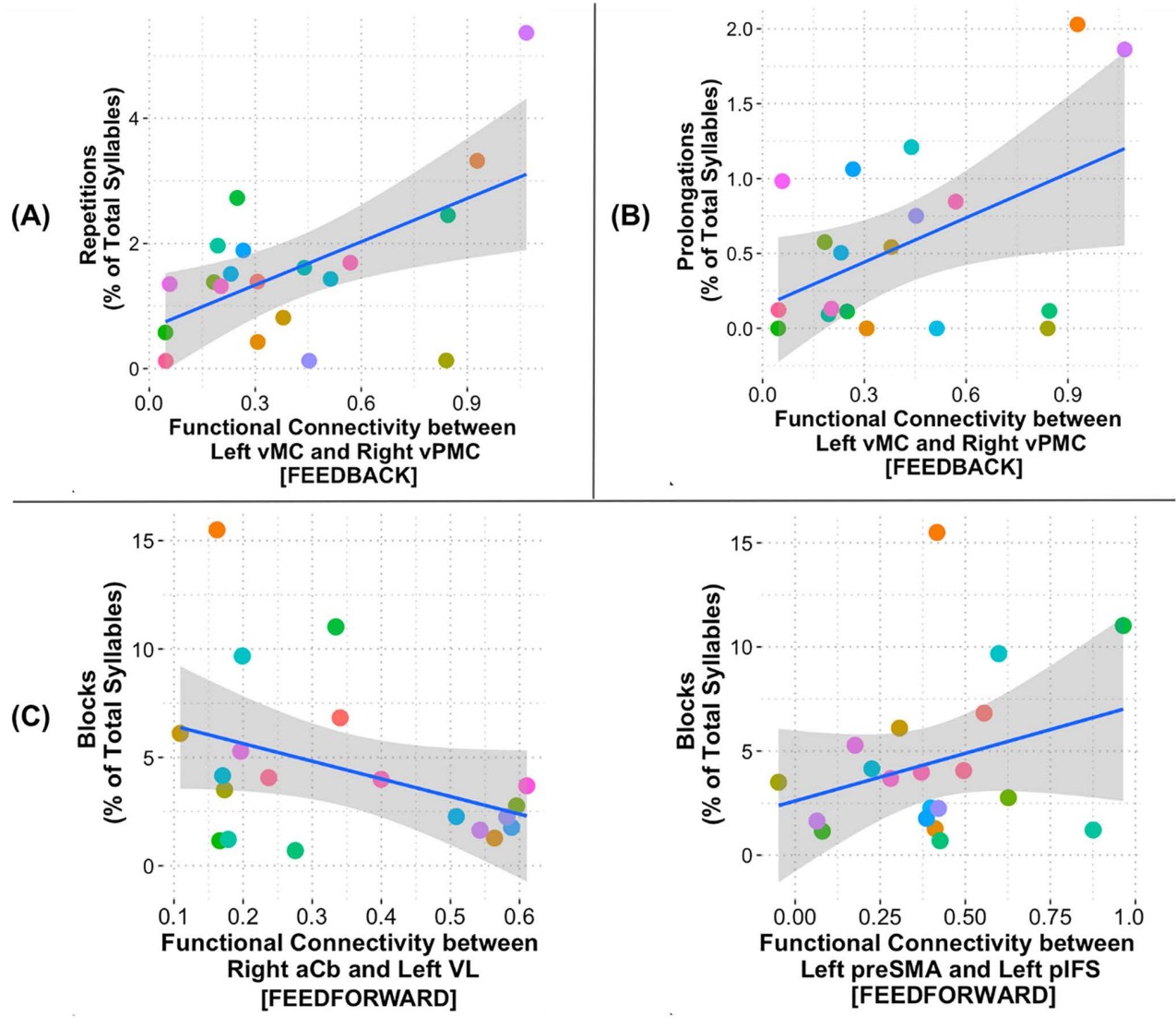

**Fig 2. Correlations between repetition rate (panel A), prolongation rate (panel B), and block rate (panel C) and functional connectivity in the feedback and feedforward control networks.** Colored dots = individual participants; aCb = anterior cerebellum; pIFS = posterior inferior frontal sulcus; preSMA = presupplementary motor area; VL = ventral lateral thalamic nucleus; vMC = ventral motor cortex; vPMC = ventral premotor cortex.

(0.008) for the left vMC to right vPMC connection suggests a strong association. For prolongations, the lower estimate (0.04) with a small standard error (0.003) for the vMC to vPMC connection indicates weaker (though still significant) association compared to repetitions. For blocks, the negative estimate (−0.18) with a relatively higher standard error (0.015) for the aCb to VL connection suggests a stronger but more variable association, while the positive estimate (0.17) with a similar standard error (0.014) for the preSMA to pIFS connection indicates a strong and consistent association. Overall, the varying lambda values reflect the different levels of complexity and regularization needed for each type of disfluency. The estimates provide insight into which connectivity patterns are most strongly associated with each disfluency type, with stronger associations seen in repetitions and blocks compared to prolongations.

## Discussion

The present study investigated the underlying neural mechanisms of different stuttering-like disfluency types (i.e., repetitions, prolongations, and blocks) using resting state functional connectivity data and a LASSO regression analysis. While we hypothesized that repetitions and prolongations may reflect greater reliance on FB-based corrective mechanisms and that blocks may stem from disruptions in FF-based initiation mechanisms, our findings suggest only partially distinct connectivity patterns across disfluency types. Rather than implying entirely separate neural pathways, the results point to overlapping yet differentially weighted contributions of FB and FF systems, which may help explain the diversity of symptom expression in AWS. See Fig 3 for the findings within the frameworks of the DIVA and GODIVA models and Fig 4 for a schematic of the brain connections associated with each disfluency type.

### Repetitions and prolongations may be associated with increased reliance on FB-based corrective mechanisms

Repetitions and prolongations were positively associated with increased connectivity between left vMC and right vPMC. This finding is in line with the prior literature suggesting a greater reliance on auditory FB for AWS [46–48] and DIVA simulations demonstrating a possible overreliance on FB control that causes the system to "reset" and repeat the motor program [59]. According to the DIVA model, a FB Control Map in right vPMC transforms sensory errors (the difference between the intended speech sound and the current sensory state) into corrective motor commands; an Articulator Map in left (and right) vMC is responsible for executing the motor commands. Therefore, increased connectivity between these regions may reflect increased reliance on FB-based corrective mechanisms. This increased reliance may interfere with the initiation of FF-based termination commands or could be a response to already disrupted timing signals in the FF control system. In either interpretation, the heightened dependence on FB-based corrective mechanisms could lead to repeated or prolonged sounds as the system continues to adjust and correct the output instead of efficiently transitioning to the next motor command. Often referred to as a hypervigilant FB control system, this account is consistent with Chang and colleagues (2019) review of the functional and neuroanatomical bases of developmental stuttering. The authors conclude that within the context of the DIVA/GODIVA framework, stuttering results from an impairment in the FF control system in the left hemisphere, which, in turn, leads to an overreliance on the FB control system in the right hemisphere [99]. More recently, in a longitudinal task-based fMRI study examining spontaneous speech in children who stutter, Chow and colleagues (2023) found that stuttering was linked to a functional anomaly in left vPMC, which extends to right premotor cortex with age. The authors concluded that right-hemisphere hyperactivation in AWS may develop later in life and, therefore, reflect unsuccessful or maladaptive compensatory behavior rather than a causal mechanism of stuttering [100].

### Blocks may be associated with disrupted subcortical timing mechanisms and continued reliance on FF-based cortical timing mechanisms

In contrast to repetitions and prolongations, blocks were negatively associated with connectivity between right aCb and left VL. This finding is consistent with the growing body of literature supporting a compensatory role of Cb in stuttering [23–31]. In the FF control system in the DIVA model, there is a direct cortico-cortical pathway (i.e., left vPMC to left vMC) and an indirect pathway that involve the right aCb to left VL connection. Based on prior work suggesting that Cb contains sensory representations that influence corrective movements [101–103], the DIVA model proposes that right Cb contributes to FF motor commands, likely through left VL, which is the main source of cerebellar input to motor cortex.

The right aCb to left VL connection is thought to play a key role in refining and optimizing motor signals prior to their initiation. While much of the supporting literature has focused on the cerebellum's role in manual motor coordination [101–103], there is growing evidence that these same predictive control mechanisms generalize to speech motor control. Similar to limb control, speech movements rely on the cerebellum to generate internal models that anticipate the sensory

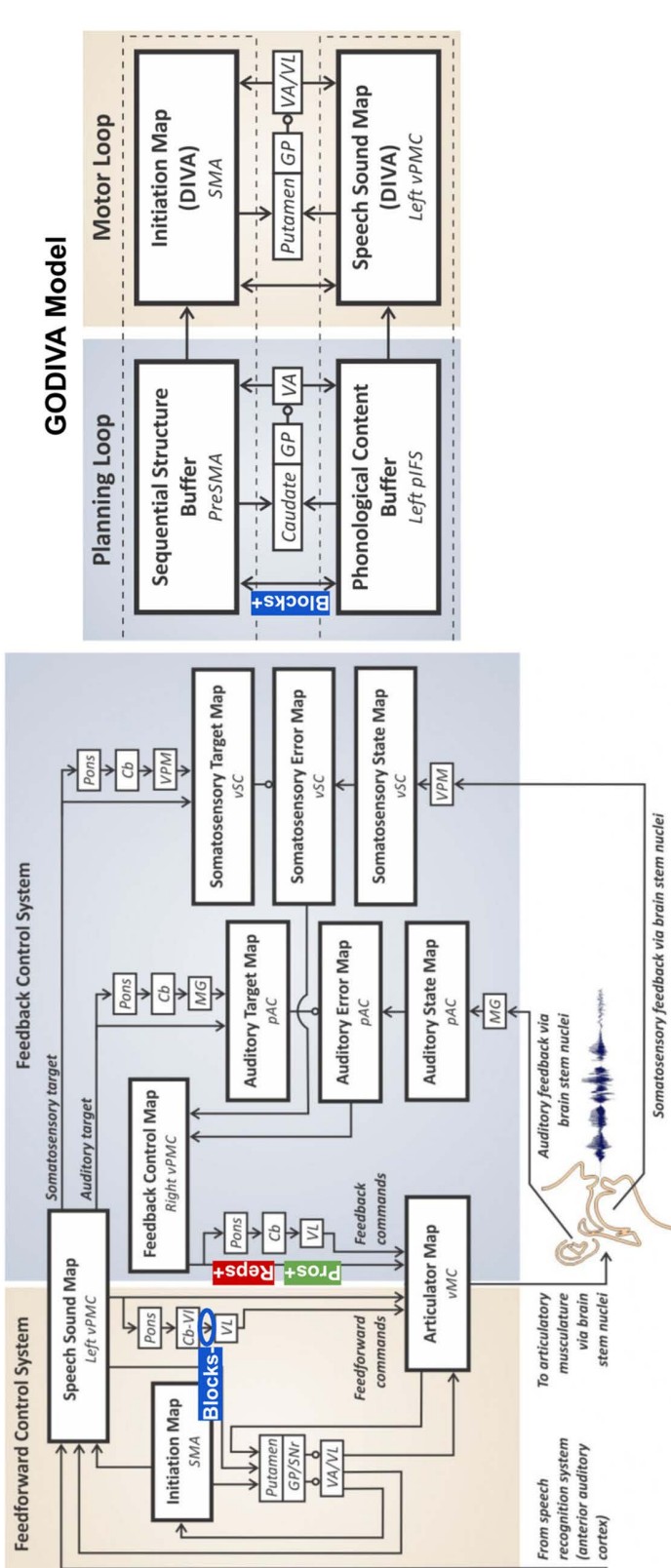

**Fig 3. Current findings within the frameworks of Directions Into Velocities of Articulators (DIVA) and Gradient Order DIVA (GODIVA).** Cb = cerebellum; GP = globus pallidus; MG = medial geniculate thalamic nucleus; pAC = posterior auditory cortex; pIFS = posterior inferior frontal sulcus; SMA = supplementary motor area; VA = ventral anterior thalamic nucleus; VL = ventral lateral thalamic nucleus; vMC = ventral motor cortex; VPM = ventral posterior medial thalamic nucleus; vPMC = ventral primary motor cortex; vSC = ventral somatosensory cortex; + = positive correlation between disfluency rate and connection strength; - = negative correlation between disfluency rate and connection strength; pros = prolongations; reps = repetitions.

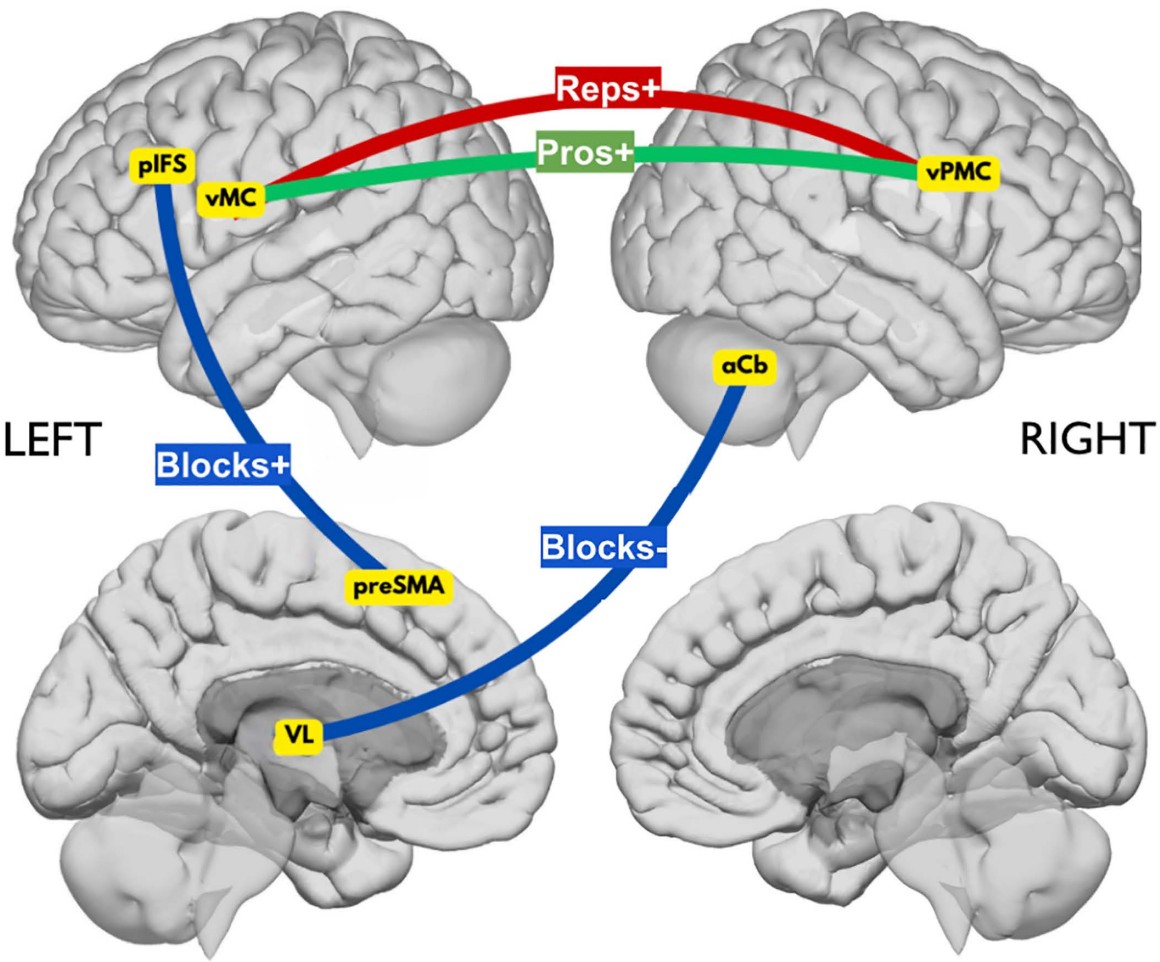

**Fig 4. Schematic of the brain connections associated with each disfluency type.** Red line = repetitions; green line = prolongations; blue line = blocks; aCb = anterior cerebellum; pIFS = posterior inferior frontal sulcus; preSMA = presupplementary motor area; VL = ventral lateral thalamic nucleus; vMC = ventral motor cortex; vPMC = ventral premotor cortex; + = positive correlation between disfluency rate and connection strength; - = negative correlation between disfluency rate and connection strength; pros = prolongations; reps = repetitions.

consequences of motor commands, allowing for the smooth and timely execution of complex motor sequences [104]. In speech, this function is particularly critical given the rapid and precise coordination required across multiple articulators. Furthermore, neuroimaging studies have shown overlapping cerebellar regions involved in both limb and speech tasks, suggesting shared computational principles [105,106]. Therefore, disruptions in cerebellar-thalamic connectivity, such as reduced connectivity between the right aCb and left VL, may reflect impairments in these predictive mechanisms, leading to difficulties initiating or coordinating speech movements. This may help explain our finding that reduced connectivity in this FF circuit is associated with an increased frequency of blocks. Of note, blocks also showed the strongest and most robust associations overall compared to repetitions and prolongations, consistent with their greater behavioral prevalence in our sample. This pattern suggests that blocks may be more tightly coupled to identifiable neural circuitry, whereas repetitions and especially prolongations may reflect more subtle or variable connectivity differences.

Blocks were also positively associated with connectivity between left pIFS and left preSMA, which are connected via the frontal aslant tract [107]. Given that preSMA projects to and is influenced by different regions of the basal ganglia and

is a critical component of the BGTC motor network [108], our findings are consistent with those of Civier and colleagues in suggesting that abnormalities in the BGTC motor network may lead to initiation difficulties associated with stuttering [67,68]. This finding also aligns with prior resting state functional connectivity studies, which demonstrated the involvement of left inferior frontal gyrus [109] and, specifically, left pIFS [110] in stuttering.

According to the GODIVA model, a Phonological Content Buffer in left pIFS stores phonological representations of intended speech sounds, while a Sequential Structure Buffer in left preSMA organizes this content into a coherent and timed sequence. When acquiring a new speech sequence, speakers initially depend on cortical structures, such as preSMA and pIFS, to guide the sequencing and initiation of sounds [53]. As learning progresses, there is a shift towards greater reliance on subcortical structures, such as basal ganglia, thalamus, and Cb [53]. Our finding of decreased connectivity between aCb and VL suggests that these subcortical structures may not fully engage in learning or fine-tuning motor commands. The failure of Cb to integrate and optimize these commands may lead to an increased dependency on cortical mechanisms. The observed association between increased connectivity between preSMA and pIFS and increased blocks may, therefore, indicate a continued reliance on cortical areas for the sequencing and initiation of sounds, reflecting a suboptimal transition to subcortical control. Thus, AWS who have more blocks due to reduced subcortical connectivity may attempt to compensate (unsuccessfully) using cortical connections. A recent structural imaging study by Neef and colleagues (2018) also found a positive association between motor signs of stuttering and frontal aslant tract connectivity strength. The authors noted that both structures coordinate with subthalamic nucleus to mediate fast global inhibition, and an overly active global response suppression mechanism could underlie the ability to smoothly execute motor actions, which may manifest as stuttering [111]. Interestingly, pIFS has also been implicated in apraxia of speech [53,112,113]—a condition that primarily affects the ability to plan and sequence the movements necessary for speech, which can manifest as blocks or prolonged pauses.

## Integrating perspectives beyond DIVA and GODIVA

While we interpreted our findings within the DIVA and GODIVA frameworks, which emphasize the role of sensorimotor integration and control mechanisms in speech production, we acknowledge that other theoretical perspectives propose alternative mechanistic drivers of specific stuttering behaviors. For example, researchers have proposed that stuttering may evolve over the lifespan, progressing from relatively effortless repetitions to more tense, rapid, or irregular repetitions, prolongations, and eventually blocks [78]. This developmental trajectory is thought to reflect increasing awareness of, and reactions to, stuttering, as well as attempts to exert control over speech, which may engage different neural substrates than those implicated by DIVA and GODIVA models. Evidence from developmental studies of stuttering further supports this view, showing age-related shifts in disfluency patterns, with older individuals tending to exhibit more blocks relative to repetitions [6].

These contrasting perspectives highlight the possibility that different stuttering behaviors—and their neural correlates—may not be fully explained by a single model. Indeed, while DIVA and GODIVA primarily focus on disruptions in FB and FF control circuits, other models emphasize emotional and cognitive responses to stuttering. For example, the Covert Repair Hypothesis [114] attributes disfluencies to internal monitoring and error correction in speech planning, while the Dual Diathesis-Stress Model [115] emphasizes how emotional reactivity, self-regulation, and cognitive load shape stuttering expression. These alternative perspectives suggest additional involvement of networks related to self-monitoring, emotional regulation, and executive control, complementing sensorimotor accounts and underscoring the multifactorial nature of stuttering. As we continue to search for the cause of stuttering, it is essential to explore how these complementary mechanisms interact, potentially integrating neural signatures associated with both sensorimotor control and higher-order cognitive-emotional processes to provide a more comprehensive account of stuttering development and variability.

## Distinct versus common underlying causes of disfluency types

Stuttering is a multifaceted disorder with variable manifestations across individuals. While this study explored the hypothesis that repetitions, prolongations, and blocks may be differentially associated with FB and FF control systems, our findings do not suggest clean distinctions between the connectivity patterns supporting these systems. Rather, the results point to considerable overlap in the neural pathways associated with different disfluency types, indicating that these control mechanisms likely interact in complex and non-exclusive ways.

Notably, it is rare for individuals to exhibit only one type of disfluency; most present with a combination of repetitions, prolongations, and blocks. This behavioral overlap is reflected in the shared involvement of several ROI connections across disfluency types, suggesting that stuttering behaviors may arise from a common set of neural mechanisms modulated by individual differences in neural dynamics or control system balance. Instead of representing entirely distinct processes, repetitions, prolongations, and blocks may reflect different manifestations of disruptions within a shared speech motor network.

Thus, this study does not argue for entirely separate neural origins of disfluency types, but rather offers an initial step toward identifying network-level variations that may help explain individual differences in stuttering expression. Future work with larger samples, symptom-specific subgrouping, and inclusion of developmental cohorts may help clarify the extent to which certain neural pathways confer vulnerability to particular disfluency profiles, and how these relationships may shift with age or developmental stage.

## Limitations

While our study provides important insights into the neural mechanisms underlying distinct speech disfluency types, it has several limitations. First, the study involved a relatively small sample size (N = 20), which may limit statistical power. In addition, the presence of zero values for each disfluency type may have influenced correlation estimates. Thus, replication with a larger, more diverse cohort would enhance the generalizability and robustness of our findings.

Second, while we limited our investigation to DIVA- and GODIVA-based connections, we acknowledge the importance of demonstrating the specificity of our findings. Follow-up studies with larger sample sizes should incorporate random or control connections to directly compare the strength and relevance of the identified connections, helping to justify the a priori selection and further substantiate the findings' specificity.

Third, our study did not account for secondary behaviors, which are often shaped by cognitive, emotional, and social factors. Incorporating measures of secondary behaviors, such as their frequency and timing relative to primary stuttering events, could help disentangle the extent to which neural activity associated with disfluencies may also reflect secondary motor responses. Additionally, studies utilizing multimodal neuroimaging and behavioral analyses could provide further insight into the interplay between motor planning disruptions and compensatory mechanisms in stuttering.

Fourth, while we examined disfluency patterns across different communication contexts, we did not model the potential influence of sample length or task duration on disfluency frequency. For example, in-person conversational samples tended to be longer than phone or reading samples, which may have increased opportunities for disfluencies to occur. Future analyses could include sample length as a covariate to better isolate effects related to context versus duration.

Fifth, although repetitions and prolongations were analyzed alongside blocks, these disfluency types made up a relatively small proportion of the total syllables produced. Given the reported variability in disfluency patterns across participants, this imbalance may limit the interpretability of neural associations specific to those less frequent disfluency types. Future studies with larger sample sizes and more balanced disfluency distributions could better support comparisons across disfluency types.

Sixth, classification of disfluency type introduces its own limitations, particularly when mixed or clustered disfluencies occur within the same moment. Although we coded by dominant disfluency type, blocks may have been more perceptible

and thus more likely to be classified as dominant, potentially underestimating the prevalence of repetitions or prolongations when they co-occurred. This perceptual weighting could influence both frequency estimates and their associated neural patterns. Moreover, debates remain about the overall utility of disfluency-type classification. Some scholars have questioned whether such measures meaningfully advance our understanding or treatment of stuttering [116,117]. These critiques underscore the need for caution in interpreting findings tied to disfluency categories and illustrate the importance of developing complementary analytic approaches (e.g., multidimensional or continuous measures of stuttering severity).

Finally, our use of resting state functional connectivity limits the ability to capture real-time neural dynamics during moments of stuttering. As such, other factors may have influenced the frequency and type of disfluencies observed that were not accounted for in our analysis. These could include individual differences in cognitive-linguistic demands, speech context, and compensatory strategies developed over time. Moreover, internal states such as anxiety, stress, attention, and linguistic proficiency likely contribute to stuttering behaviors and may modulate neural network activity in ways not captured by resting-state measures. Future studies would benefit from incorporating direct assessments of these individual factors to better contextualize the neural mechanisms associated with disfluency patterns and improve the interpretability of resting-state connectivity findings. Additionally, task-based fMRI (e.g., controlled elicitation of stuttering behaviors; [118]), real-time neural recordings during speech, and comprehensive behavioral assessments could help disentangle these complex interactions. Longitudinal and task-based designs may further clarify the dynamic relationships between neural connectivity and situational influences, ultimately providing a more comprehensive understanding of the mechanisms underlying speech disfluencies.

## Theoretical and clinical implications

Overall, our findings provide evidence of partially distinct yet overlapping mechanisms underlying different stuttering behaviors, which has important theoretical implications for models of speech production and motor control. Specifically, these results support frameworks that differentiate between FB- and FF-based speech regulation, while also highlighting shared neural pathways. This view challenges the notion of a singular neural deficit in stuttering and instead points to a more nuanced, mechanism-specific perspective. Additionally, these findings motivate the need for theoretical models to account for individual variability in neural control strategies, which may inform more personalized approaches to understanding and treating stuttering.

From a clinical standpoint, further research in this area has the potential to inform the development of more targeted behavioral and medical interventions for individuals who stutter, ultimately improving treatment efficacy and personalization. For example, if repetitions and prolongations are associated with an overreliance on auditory FB mechanisms, behavioral strategies targeting enhanced control of auditory feedback, such as those using speech feedback manipulation (e.g., delayed auditory feedback), could benefit individuals exhibiting these disfluency types. Similarly, if blocks are associated with difficulties in initiating speech due to disruptions in the FF system and subcortical timing mechanisms, treatment strategies focusing on enhanced speech initiation or smoother transitions between speech motor programs could be more effective for individuals who primarily exhibit blocks.

Additional research could also inform future medical interventions (e.g., neuromodulation or pharmacological treatments) targeting specific neural pathways. Prior studies have reported increased fluency following stimulation to speech motor areas, including inferior frontal gyrus and supplementary motor area [119,120]. Similarly, pharmacological interventions have targeted dopaminergic pathways, given the proposed role of dopamine dysregulation in stuttering [121,122]. Building on these findings, individuals with FB-based disfluencies (e.g., repetitions, prolongations) may benefit from neuromodulation to the left vMC to right vPMC pathway, with the goal of recalibrating an overactive feedback system. For individuals with blocks, which may be linked to timing disruptions in subcortical circuits, pharmacological treatments targeting the basal ganglia or cerebellum could offer more tailored approaches to restore motor initiation and timing function.

In conclusion, these findings underscore the potential for developing personalized, targeted interventions that consider the different neural mechanisms involved in various stuttering behaviors. By targeting the underlying neural mechanisms associated with different disfluency types, both behavioral and medical treatments for stuttering could be made more effective and personalized. Further research with larger sample sizes will provide stronger evidence to guide the development of future behavioral and medical interventions, helping to identify the most effective targets for treatment based on specific neural pathways associated with different disfluency types.

## Supporting information

**Table S1. Results of Levene's tests assessing potential heteroscedasticity due to inter-site variance.**
(DOCX)

**Table S2. Coefficient estimates, standard errors, and 95% confidence intervals of connections with statistically significant effects identified by each LASSO regression.**
(DOCX)

**Fig S1. Selection frequencies plotted in decreasing order.** Red point = cutoff used to distinguish stable from unstable predictors, identified as the last variable before the largest drop in selection frequency; aCb = anterior cerebellum; pIFS = posterior inferior frontal sulcus; preSMA = presupplementary motor area; VL = ventral lateral thalamic nucleus; vMC = ventral motor cortex; vPMC = ventral premotor cortex.
(TIF)

## Acknowledgments

We thank all participants for their time and contributions to this research.

## Author contributions

**Conceptualization:** Hannah P. Rowe, Jason A. Tourville, Frank H. Guenther.

**Data curation:** Saul A. Frankford, Jason A. Tourville.

**Formal analysis:** Hannah P. Rowe, Saul A. Frankford, Jason A. Tourville.

**Funding acquisition:** Frank H. Guenther.

**Investigation:** Saul A. Frankford, Jackie S. Kim.

**Methodology:** Hannah P. Rowe, Saul A. Frankford, Jackie S. Kim, Jason A. Tourville, Frank H. Guenther.

**Project administration:** Frank H. Guenther.

**Resources:** Frank H. Guenther.

**Software:** Jason A. Tourville, Alfonso Nieto-Castanon.

**Supervision:** Jason A. Tourville, Frank H. Guenther.

**Validation:** Alfonso Nieto-Castanon.

**Visualization:** Hannah P. Rowe.

**Writing – original draft:** Hannah P. Rowe.

**Writing – review & editing:** Saul A. Frankford, Jackie S. Kim, Jason A. Tourville, Alfonso Nieto-Castanon, Frank H. Guenther.

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
