## [Decision Letter · Decision Letter 0]

31 Jul 2025

Dear Dr. Rowe,

Thank you for submitting your manuscript to PLOS ONE. After careful consideration, we feel that it has merit but does not fully meet PLOS ONE’s publication criteria as it currently stands. Therefore, we invite you to submit a revised version of the manuscript that addresses the points raised during the review process.

We look forward to receiving your revised manuscript.

Kind regards,

Li-Hsin Ning

Academic Editor

PLOS ONE

“This work was supported by the National Institutes of Health (National Institute of Deafness and Other Communication Disorders, https://www.nidcd.nih.gov/) [FHG: R01DC007683 and HPR: T32DC013017-07].”

“This work was supported by the National Institutes of Health [grant numbers R01DC007683 and T32DC013017-07].”

“This work was supported by the National Institutes of Health (National Institute of Deafness and Other Communication Disorders, https://www.nidcd.nih.gov/) [FHG: R01DC007683 and HPR: T32DC013017-07].”

“Frank Guenther receives royalties for his book Neural Control of Speech from MIT Press. All other authors have no known competing financial interests or personal relationships that could have appeared to influence the work reported in this paper.”

5. We note that you have indicated that there are restrictions to data sharing for this study. PLOS only allows data to be available upon request if there are legal or ethical restrictions on sharing data publicly. For more information on unacceptable data access restrictions, please see http://journals.plos.org/plosone/s/data-availability#loc-unacceptable-data-access-restrictions.

Additional Editor Comments:

The manuscript investigates the neural correlates of FB control and FF control across three types of stuttering (repetitions, prolongations, and blocks). While I found the manuscript to be interesting and well-organized, I have several clarification questions that I hope the authors can address in the revised manuscript.

Please add page numbers and line numbers.

Table 1: Please elaborate on how the SSI-4 was measured for each participant. In addition to the percentages, please also provide the raw counts and the total number of syllables for each cell.

Sect 2.2, Para 1, Line 4: “The topic of the conversational samples” -> Make it clear that this is “in-person conversational samples”

Sect 2.2, Para 1: Your in-person conversational samples were, on average, twice as long as the phone conversational samples and the reading samples. I wonder why the task effect was not tested in the statistical model. Could it be that people who stutter tend to produce more stuttering behaviors as speech production length increases?

Sect 2.2, Para 2, Line 8: “…involves the occurrence of two or more different stuttering types in a single instance” -> What exactly do you mean by “single instance”? Is it a sentence, an utterance, a phrase, a word, or something else?

Sect 2.2, Para 2, Lines 9-10: “…the stutter was classified according to the dominant disfluency type within that occurrence” -> What if there was a tie between disfluency types? How was the category assigned in that case?

Sect 2.5, Para 2, Lines 9-10: “The results did not indicate a significant effect, suggesting that further normalization across sites was not necessary.” -> Please provide the Levene’s test statistics, as these are required to support this claim.

Sect 3.2, Para 1, Lines 7-9: “All predictors showed statistically significant effects, as their CIs did not include zero, indicating consistent and non-random relationships across the resampled datasets.” -> Please provide a summary table or a forest plot to support this claim.

Figure 1 caption: You may need to explain what the black rectangles (the selected variables after applying the elbow method?) represent, both in the figure caption and in the main text.

Sect 3.2, Para 2, Lines 1-4: “The stability selection method was then conducted to identify the most consistently influential predictors across the 1000 iterations. Lastly, the elbow method was applied to determine the cutoff point for the predictors that had non-zero coefficients in the highest number of model iterations.” -> Please provide the elbow plot for each type (repetition, prolongation, and block).

Sect 4.2, Para 4 and Figure 4: There is a phonological content buffer in the planning loop of the GODIVA model. I’m curious whether the participants had more difficulty initiating onset consonants or the following vowels (e.g., in a CV syllable). Given that vowels are generally more prominent or dominant in a syllable, did they experience more blocks during vowel production compared to consonant production?

Please check the apostrophes and quotes throughout the entire manuscript, as some appear to have inconsistent formatting (possibly from copying and pasting from other resources). For example, see Line 8 of Section 1.4, Lines 8 and 10 of Section 2.3, Line 10 of Section 2.4.2, and Line 10 of Section 5.

Reviewers' comments:

Reviewer's Responses to Questions

**Comments to the Author**

1. Is the manuscript technically sound, and do the data support the conclusions?

Reviewer #1: Partly

2. Has the statistical analysis been performed appropriately and rigorously?

Reviewer #1: Yes

3. Have the authors made all data underlying the findings in their manuscript fully available?

Reviewer #1: No

4. Is the manuscript presented in an intelligible fashion and written in standard English?

Reviewer #1: Yes

Reviewer #1: This paper reports on the association between the overt speech characteristics of stuttering, involving repetitions, prolongations, and blocks, and resting state functional connectivity among adults who stutter. The authors describe their findings of associations between repetitions and prolongations and increased connectivity between left ventral motor and right ventral premotor cortices, as well as associations between blocks and connectivity between right anterior cerebellum and ventral lateral thalamus in the context of the feedback and feedforward control mechanisms that are theoretically modeled by the DIVA framework and have also been empirically investigated.

Although a modest sample of adults who stutter is included, the authors have applied LASSO regression to try to address potential biases and variance in their analysis of heterogenous resting state functional connectivity data.

The rationale for this investigation is well placed within the context of previous research showing evidence of heterogeneity along with defined behavioural mechanisms and neural distinctions that are differentially represented among persons who stutter. The methods are well described and appear sound.

There are, however, some points that I have raised below, which I hope can guide the authors in improving the clarity of their reporting for the reader.

1. Introduction:

- In line with the appropriate reference to the heterogeneity of the stuttering experience and its overt presentation, can the authors provide justification for their claims in determining the mechanisms affected among the subtypes of adults who stutter in this study, and whether this study was successful in contributing to a more unified understanding of stuttering?

- In the Introduction, please elaborate on the value of the proposed mechanistic understanding of stuttering to the success of treatment management and outcomes.

- The authors may consider revising what is deemed as the typical characterization of stuttering, and what are overt speech characteristics.

- Please consider also revising the use of the term “abnormalities” in reference to feedback and feedforward processing, with distinctions or differential behavioural strategies associated with neurodevelopmental differences in stuttering.

- Consider also revising the use of the term “controls” in section 1.2.

- In the second sentence of section 1.4, please clarify that the resting state functional connectivity patterns of persons with persistent stuttering are being investigated.

- While describing the advantages of analysing resting state functional connectivity patterns, it would be helpful if the authors also outline the potential disadvantages, if any, when task-based functional activation in persons who stutter is interpreted.

- When stating that the hypotheses regarding feedback and feedforward control are based on the aforementioned literature, please recite these specific studies for the reader.

- In addition, when linking different stuttering behaviors to the feedback and feedforward control networks, could the authors specify whether they expected network connectivity to be increased or decreased?

- The authors state that these findings could guide future research. Could they specify potential areas of further investigation?

2. Methods:

- Was clinical diagnosis conducted in the classification of these participants, or did this rely on self-disclosure alone? If the latter, the limitations and potential confound of this should be addressed.

- For Table 1, please explain each of the abbreviations used within the table, as well as revising the punctuation in the first sentence of the caption, and stating the total number of syllables elicited during speech testing.

- Please state the literature used to categorize the stuttering behaviors and guide the coding procedures.

- Please state the training and/or qualifications of the raters included in the reliability analysis.

- In section 2.4, please cite the validation studies for the CONN toolbox, as well as referenced support for the application of both surface-based and volumetric approaches for the analysis of functional connectivity between ROIs.

- Could the authors please also cite support for their approach to bootstrapping?

3. Results:

- With repetitions and prolongations making up such a small proportion of the syllables used for analysis, along with the reported variability of all disfluency types among participants, could the authors speak to the limitations of this in interpreting their results?

- Figure 1: By visual inspection, it appears that the left medial geniculate thalamic nucleus and left Heschl’s gyrus show similar magnitude for blocks to that for prolongations. Can the authors report on the significance of this association and clarify why this is not discussed? This may speak to the authors’ recognition that individual differences or heterogeneity of resting state functional connectivity may be limited in providing unique or distinct neural characteristics of developmental stuttering.

4. Discussion

- The literature that is cited to support the contribution of the anterior cerebellum in optimizing the accuracy of speech motor commands is primarily focused on manual motor coordination. Can the authors clarify this within the text and speak to the generalizability of these previous research findings to speech motor coordination?

- Section 4.4 highlights the fact that not one type of stuttering behavior is specific to an individual’s experience and presentation of stuttering, with the overlap of associations between ROI connectivity also warranting further emphasis.

- In discussing the limitations of using resting state functional connectivity, the authors may consider further highlighting the need to measure individuals’ experiences of anxiety, stress, attention, and linguistic proficiency.

5. Theoretical and Clinical Implications

- It is important to note that the findings do not cleanly distinguish or differentiate the connectivity patterns that support feedback and feedforward control. The interpretation of results may be more nuanced than the authors have described.

- In discussing the potential benefits of targeted neuromodulation or pharmacology, could the authors briefly summarize the findings from studies of neuromodulatory and pharmacological effects in this population?

I hope that my suggestions can help that authors refine their reporting and clarify the complexities of interpreting the speech motor and neural circuitry characteristics of stuttering to the reader. I commend their efforts in deciphering these highly variable stuttering behaviors and relating them to the neural characteristics that define this neurodevelopmental condition.

**Do you want your identity to be public for this peer review?** For information about this choice, including consent withdrawal, please see our Privacy Policy

Reviewer #1: **Yes: ** Fiona Höbler

---

## [Author Response · Author response to Decision Letter 1]

6 Aug 2025

Please see my Response Letter (attached).

---

## [Decision Letter · Decision Letter 1]

3 Sep 2025

Dear Dr. Rowe,

Thank you for submitting your manuscript to PLOS ONE. After careful consideration, we feel that it has merit but does not fully meet PLOS ONE’s publication criteria as it currently stands. Therefore, we invite you to submit a revised version of the manuscript that addresses the points raised during the review process.

We look forward to receiving your revised manuscript.

Kind regards,

Li-Hsin Ning

Academic Editor

PLOS ONE

Journal Requirements:

Reviewers' comments:

Reviewer's Responses to Questions

**Comments to the Author**

Reviewer #1: (No Response)

2. Is the manuscript technically sound, and do the data support the conclusions?

Reviewer #1: Yes

3. Has the statistical analysis been performed appropriately and rigorously?

Reviewer #1: Yes

4. Have the authors made all data underlying the findings in their manuscript fully available?

Reviewer #1: Yes

5. Is the manuscript presented in an intelligible fashion and written in standard English?

Reviewer #1: Yes

Reviewer #1: The authors have addressed the main points previously raised in their revised manuscript. A few minor issues remain, which I hope can further improve the reporting of this investigation into disfluency types and their underlying neural mechanisms in adults who stutter. These issues mostly concern the reporting of identified neural associations as being distinct, when overlap of associated connections was found, as well as the categorization of disfluency types and the predominance of blocks identified in the speech of study participants.

Abstract:

• Page 2, line 17: The authors note in their discussion of results that their “findings do not suggest clean distinctions between the connectivity patterns supporting these systems”. Yet, in the Abstract, they report the identified neural underpinnings as being “distinct”. For consistency, please revise this interpretation of distinctness throughout the manuscript, including the later section on Theoretical and clinical implications.

Introduction

• Page 3, lines 2-3: In the interest of recognizing the multifaceted experience of stuttering, the authors may consider describing the “overt speech behaviors” of stuttering as being generally characterized by repetitions, prolongations and blocks.

• Page 7, line 15: The authors may wish to vary the wording for “capturing” in this sentence, which appears twice.

Methods

• Page 9, line 4: If interpreted correctly, the term “social context” may need to be pluralized here.

• Page 11, line 12: It is still unclear how moments of disfluency were classified according to the dominant disfluency type. Was this dominance judged by the first disfluency type to occur or measured by the length of the disfluency type? The issue of mixed disfluencies may need to be addressed here, as well as how the potential dominance of perceptible blocks may have also influenced their prevalence within participants’ speech samples. This is again noted below.

• Page 12, line 23: With two citations referenced in support of CONN toolbox validation, the authors may wish to note that this has been “validated in the field” and not “extensively”, unless specific validation studies can be cited.

Discussion

• Page 26, lines 4-9: Here it is stated that “researchers have proposed that stuttering may evolve over the lifespan, progressing from relatively effortless repetitions to more tense, rapid, or irregular repetitions, prolongations, and eventually blocks” which may reflect “an increasing awareness of, and reactions to, stuttering, as well as attempts to exert control over speech”. Can the authors identify additional support of these claims in the literature? The following report on age effects on disfluency types may be considered here: Staróbole Juste, F., & Furquim de Andrade, C. R. (2011). Speech disfluency types of fluent and stuttering individuals: age effects. Folia Phoniatrica et Logopaedica, 63(2), 57-64.

• Page 26, lines 10-14: Please state which models emphasize the emotional and cognitive responses to stuttering. References to alternative and/or complimentary models, frameworks and/or theories should be included here.

• Pages 26-27, lines 22-10: As mentioned earlier, the authors’ interpretation that “repetitions, prolongations, and blocks may reflect different manifestations of disruptions within a shared speech motor network”, as opposed to distinct neural mechanisms, could be more consistently reported throughout the manuscript.

• Page 27, lines 13-15: In considering potential follow-up investigations and future research on this topic, potential age or developmental effects on disfluency types, their neural associations, and their predominance could be listed here. In addition, the limitations of disfluency type classification should be considered, particularly when mixed or clustered disfluencies arise. In the current investigation, consideration could also be given to how the disfluency type of “blocks” may be more perceptible or perceived as being more dominant, consequently reducing the perceived prevalence of repetitions or prolongations – particularly if these behaviors were more subtle. Arguments against disfluency-type classification have also been made:

o Einarsdóttir, J., & Ingham, R. J. (2005). Have disfluency-type measures contributed to the understanding and treatment of developmental stuttering?. American Journal of Speech-Language Pathology, 14(4), 260-273.

o Bothe, A. K. (2008). Identification of children’s stuttered and nonstuttered speech by highly experienced judges: Binary judgments and comparisons with disfluency-types definitions. Journal of Speech, Language, and Hearing Research, 51(4), 867-878.

Supplementary materials:

• Table S2: It appears that blocks had the most statistically significant effects, followed by repetitions, and prolongations having fewer statistically significant effects on resting state connections. The relevance of these findings could be discussed within the reporting of results, as well as their potential implications in the discussion section.

I hope that these suggestions are helpful to the authors in their reporting of this work.

**Do you want your identity to be public for this peer review?** For information about this choice, including consent withdrawal, please see our Privacy Policy

Reviewer #1: **Yes: **

---

## [Author Response · Author response to Decision Letter 2]

4 Sep 2025

Please see attached Response Letter.

---

## [Editor Report · Decision Letter 2]

10 Sep 2025

Differential involvement of feedback and feedforward control networks across disfluency types in adults who stutter: Evidence from resting state functional connectivity

PONE-D-25-15878R2

Dear Dr. Rowe,

We’re pleased to inform you that your manuscript has been judged scientifically suitable for publication and will be formally accepted for publication once it meets all outstanding technical requirements.

Kind regards,

Li-Hsin Ning

Academic Editor

PLOS ONE
---

## [Editor Report · Acceptance letter]

PONE-D-25-15878R2

PLOS ONE

Dear Dr. Rowe,

I'm pleased to inform you that your manuscript has been deemed suitable for publication in PLOS ONE. Congratulations! Your manuscript is now being handed over to our production team.

Kind regards,

on behalf of

Dr. Li-Hsin Ning

Academic Editor

PLOS ONE